# STATISTICAL PROPERTY TESTING FOR GENERATIVE MODELS

**Emmanouil Seferis, Simon Burton, Chih-Hong Cheng**
Fraunhofer Institute for Cognitive Systems (IKS)
Hansastr. 32, Munich, Germany
{emmanouil.seferis,simon.burton,chih-hong.cheng}@iks.fraunhofer.de

## ABSTRACT

Generative models that produce images, text, or other types of data are recently be equipped with more powerful capabilities. Nevertheless, in some use cases of the generated data (e.g., using it for model training), one must ensure that the synthetic data points satisfy some properties that make them suitable for the intended use. Towards this goal, we present a simple framework to statistically check if the data produced by a generative model satisfy some property with a given confidence level. We apply our methodology to standard image and text-to-image generative models.

## 1 INTRODUCTION & RELATED WORK

Generative models have achieved remarkable progress in the past few years. First, models such as Generative Adversarial Networks (GANs) Goodfellow et al. (2020) or Variational Autoencoders (VAE) Kingma et al. (2019) have managed to create artificial images whose faithfulness can not be distinguished by human vision. Moreover, there have been extensions of GANs and VAEs for more specific use cases, such as creating synthetic data of given sorts, e.g. generating images with given styles, among others. More recently, Diffusion Models Ho et al. (2020) have matched or surpassed GANs in terms of performance. Beyond images, very recently, multi-modal generative models have been developed, for example for NLP applications Brown et al. (2020) or for creating artificial images based on text prompts Ramesh et al. (2021); Rombach et al. (2021).

Nevertheless, depending on the use-case, it might be necessary to make sure that the generated data satisfies some given properties of interest. For example, in the case where we wish to use synthetic data for training other machine learning models on them, we need to ensure that the generated data is realistic. In this paper, based on combining hypothesis testing and the Hoeffding's inequality, we present a simple framework in order to test that the data produced by a generative model satisfy some given property at a specified confidence level. We showcase our method for images generated by GANs, as well as for text to image generative models.

In terms of related work, there is a growing body of work concerned about the safety properties of Deep Neural Networks (DNNs), for example, from the perspective of formal methods Huang et al. (2017); Cheng et al. (2020) or statistics Angelopoulos & Bates (2021). However, most of these works focus on deriving safety claims from discriminative models. For safety assurance of discriminative models, assumptions on the underlying data must be made, such as the data being collected in an i.i.d. (independently and identically distributed) manner and the data covering the whole operational domain. These assumptions are difficult to verify, as the real-world data distribution is often unknown. This is in contrast to the case of generative models, where the data distribution is well defined (e.g., the distribution being characterized by a DNN), and therefore sound statistical claims can be derived.

## 2 METHODOLOGY

Let $G$ be a generative model that outputs data points $\mathbf{x} \in \mathbb{R}^d$ based on a distribution $p_G(.)$; we slightly abuse notation and write $\mathbf{x} \sim G$. For example, $\mathbf{x}$ could represent images, text tokens,

Table 1: Results: In experiment 1, we test property $\phi_1$ on the GTSRB GAN. We generate $n = 500$ stop signs, out of which 351 are correctly recognized. In experiment 2, we test property $\phi_2$ on SD. (a) In the 1st case, the results for the easier prompts are shown; we generated $n_1 = 300$ prompts, and SD failed to generate a car in 12 cases. (b) In the 2nd experiments we have the results for the harder prompts; we generated $n_1 = 200$ prompts, and SD failed to generate a car in 52 cases.

| Experiment: | Hypothesis $H_0$ | $p_0 = 50\%$ | $p_0 = 60\%$ | $p_0 = 70\%$ | $p_0 = 80\%$ | $p_0 = 90\%$ |
|---|---|---|---|---|---|---|
| Experiment 1 | confidence $1 - \delta$ | 1.00 | 0.99 | 0.00 | - | - |
| Experiment 2 (a) | confidence $1 - \delta$ | 1.00 | 1.00 | 1.00 | 0.99 | 0.88 |
| Experiment 2 (b) | confidence $1 - \delta$ | 1.00 | 0.99 | 0.42 | - | - |

or other kinds of data. For some generative models (for example text to image), the generated samples are conditional to a given prompt, $\mathbf{c} \in \mathbb{R}^k$, and in that case we write $\mathbf{x} \sim G(\mathbf{c})$, with the understanding that the data $\mathbf{x}$ follow a distribution $p_G(.|\mathbf{c})$ conditional to $\mathbf{c}$.

Further, let $\phi : \mathbb{R}^d \to \{0, 1\}$ be a property that we want to verify. We imagine $\phi$ as a function, that, upon input of a given $\mathbf{x}$, returns 1 (true) or 0 (false), depending whether the property is satisfied or not. In practice, $\phi$ can be implemented by an oracle such as human inspection, or by a discriminative model that classifies whether its input $\mathbf{x}$ satisfies the property or not.

We want to test the hypothesis that $G$ outputs data that satisfy the property with sufficiently high probability. For this, we formulate the (null) hypothesis $H_0 = \{p \geq p_0\}$, where $p = P_{\mathbf{x} \sim G}[\phi(\mathbf{x}) = 1]$ is the probability that $G$'s outputs satisfy the property, and $p_0$ is a required lower bound that we want to achieve. The goal is then to verify that $H_0$ holds with sufficiently high confidence. From Hoeffding's inequality, we can easily derive the following:

**Proposition 1**: Let $\mathbf{x}_1, \ldots, \mathbf{x}_n$ be $n$ i.i.d. samples from $G$, and put $y_i = \phi(\mathbf{x}_i), i = 1, \ldots, n$. Further, consider the proportion $\bar{S}_n = (y_1 + \ldots + y_n)/n$. If $\bar{S}_n \geq p_0$, then the null hypothesis $H_0 = \{p \geq p_0\}$ (as defined above) holds with confidence of at least $1 - \delta$, where $\delta = \exp(-2n(\bar{S}_n - p_0)^2)$.

In the case of conditional generative models, we want to test whether the model's outputs satisfy the given property $\phi$ (with sufficiently high confidence), when the prompts also satisfy the property. For example, we may wish to test whether the generated images contain cars, provided that the prompts are descriptions of cars. To this end, our goal is to test the conditional probability $p_G(.|\mathbf{c})$ where prompts $\mathbf{c}$ comply with $\phi$. For that, Proposition 1 can be used as well.

## 3 EXPERIMENTS

In our first experiment, we generated artificial traffic sign images using the GAN of Spata et al. (2019) that intend to look similar to the GTSRB dataset Stallkamp et al. (2011). We generated synthetic stop signs, and aim to test whether the synthetic images are indeed stop signs. For that, as our property $\phi_1$ we used a custom Convolutional Neural Network (CNN) that achieves $\geq 95\%$ accuracy on the GTSRB test set. For various confidence levels, Table 1 details the results.

For our second experiment, we worked with the Stable Diffusion (SD) Rombach et al. (2021) text-to-image generative model. We created $n_1 = 300$ short prompts describing a car (e.g. "A self-driving car navigating a busy intersection."), and tested the property $\phi_2$ on whether the generated image indeed contains a car. To test $\phi_2$, we use an object detection model (YOLOv8 Jocher et al. (2023)) and return true if it detects a car. Further, we conducted a second round of the experiment, where we create $n_2 = 200$ longer and more complicated prompts. The results are also shown in Table 1.

## 4 CONCLUSION

In this work, we discussed how to statistically test the properties of data created by generative models to obtain sound confidence levels. Beyond the presented approach, many directions for future work are possible. For example, it is unclear which properties synthetic data should satisfy if it is to be used for subsequent model training. Another interesting avenue is how one can train generative models in order to satisfy a given set of properties with high confidence.

URM STATEMENT

We acknowledge that all authors of this work meet the URM criteria of ICLR 2023 Tiny Papers Track.

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

# A    APPENDIX

## A.1    DEFERRED PROOFS

**Proposition 1**: Let $\mathbf{x}_1, \ldots, \mathbf{x}_n$ be $n$ i.i.d. samples from $G$, and put $y_i = \phi(\mathbf{x}_i), i = 1, \ldots, n$. Further, consider the proportion $\bar{S}_n = (y_1 + \ldots + y_n)/n$. If $\bar{S}_n \geq p_0$, then the null hypothesis $H_0 = \{p \geq p_0\}$ (as defined above) holds with confidence of at least $1 - \delta$, where $\delta = \exp(-2n(\bar{S}_n - p_0)^2)$.

*Proof (Proposition 1)*:   From Hoeffding's inequality Shalev-Shwartz & Ben-David (2014), if $Y_1, \ldots, Y_n$ are independent random variables with $a_i \leq Y_i \leq b_i$ almost surely, then, setting $S_n = Y_1 + \ldots + Y_n$ we have, for all $t > 0$:

$$\mathbb{P}[S_n - \mathbb{E}[S_n] \geq t] \leq \exp\left[-\frac{2t^2}{\sum_{i=1}^n (b_i - a_i)^2}\right] \tag{1}$$

In our case, the random variables $Y_i = \phi(X_i), X_i \sim G$ are Bernoulli and bounded in $[0, 1]$, hence $a_i = 0, b_i = 1$ and $\sum_{i=1}^n (b_i - a_i)^2 = n$, so eq. 1 becomes:

$$\mathbb{P}[S_n - \mathbb{E}[S_n] \geq t] \leq \exp\left[-\frac{2t^2}{n}\right] \tag{2}$$

Now, we introduce the ratio $\bar{S}_n = S_n/n$, and $\mathbb{E}[\bar{S}_n] = \mathbb{E}[S_n]/n$ respectively. With these, the probability statement on the left-hand side of eq. 2 can be written equivalently as:

$$S_n - \mathbb{E}[S_n] \geq t \iff \bar{S}_n - \mathbb{E}[\bar{S}_n] \geq t/n \tag{3}$$

Plugging 3 in 2, setting $t/n = \epsilon$ and solving for $\epsilon$, we get:

$$\mathbb{P}[\bar{S}_n - \mathbb{E}[\bar{S}_n] \geq \epsilon] \leq \exp[-2n\epsilon^2] \tag{4}$$

Next, since $Y_i$ are Bernoullis with (unknown) probability $p$, we have that $\mathbb{E}[\bar{S}_n] = p$, which gives us:

$$\mathbb{P}[\bar{S}_n - p \geq \epsilon] \leq \exp[-2n\epsilon^2] \tag{5}$$

Finally, write the statement inside the probability equivalently as $p \leq \bar{S}_n - \epsilon$, and set $\bar{S}_n - \epsilon = p_0$, which gives:

$$\mathbb{P}[p \leq p_0] \leq \exp[-2n\epsilon^2] \tag{6}$$

Finally, eliminate $\epsilon$ on the right-hand side by replacing it with $\epsilon = \bar{S}_n - p_0$, and call the right-hand side $\delta$. This gives:

$$\mathbb{P}[p \leq p_0] \leq \exp[-2n(\bar{S}_n - p_0)^2] := \delta \tag{7}$$

Therefore, for the opposite statement, we get $\mathbb{P}[p \geq p_0] \geq 1 - \delta$, as required.    $\square$

## A.2    EXPERIMENTAL DETAILS

In this section we present some further experimental details, omitted in the main text.

For experiment 1, we trained a CNN with the following architecture to recognize traffic signs:

ReLU(BN(CONV(40))) $\rightarrow$ MaxPool $\rightarrow$ ReLU(BN(CONV(20))) $\rightarrow$ MaxPool $\rightarrow$ ReLU(fc(240)) $\rightarrow$ ReLU(fc(84)) $\rightarrow$ fc(43)

Convolutional layers (CONV) have (5,5) kernel and stride 1, while Max Pooling (MaxPool) layers are $2 \times 2$. fc denotes fully-connected layers, BN batch normalization, and ReLU is the ReLU activation function. We train this CNN with the Adam optimizer for 10 epochs and learning rate $10^{-3}$, until it achieves an accuracy $96\%$ on the GTSRB test-set.

For experiment 2, we used SD version 1.4 from the official repository. For the prompts, we generate them with ChatGPT Brown et al. (2020) and then inspect them manually, to check that they are all correct by containing descriptions of cars.

In experiment 2a, we generated short prompts of a few words, such as the examples below:

- "A red sports car speeding down a coastal highway."
- "A silver Porsche driving through a snowy landscape."
- "A red Audi sports car driving through the countryside."
- "A black Ford F-150 parked in front of a construction site."

In experiment 2b, we generated longer prompts that intend to be more complicated and contain several objects (besides cars). The following showcases some texts generated:

- "Two cars driving on a narrow mountain road with sharp turns, with a mountain biker in the middle of the road."
- "A car driving through a city during a carnival, with elaborate costumes and floats parading down the street."
- "A car driving through a city during a technology conference, with keynote speakers and tech demos."
- "A car driving through a small town during a peach festival, with peach pie and other peach-themed treats."

Below we showed the generated images for the above randomly selected prompts.

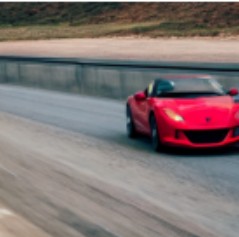 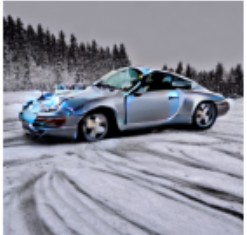
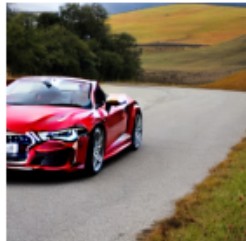 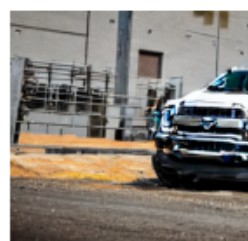

Figure 1: Generated images for the above random prompts from experiment 2a.

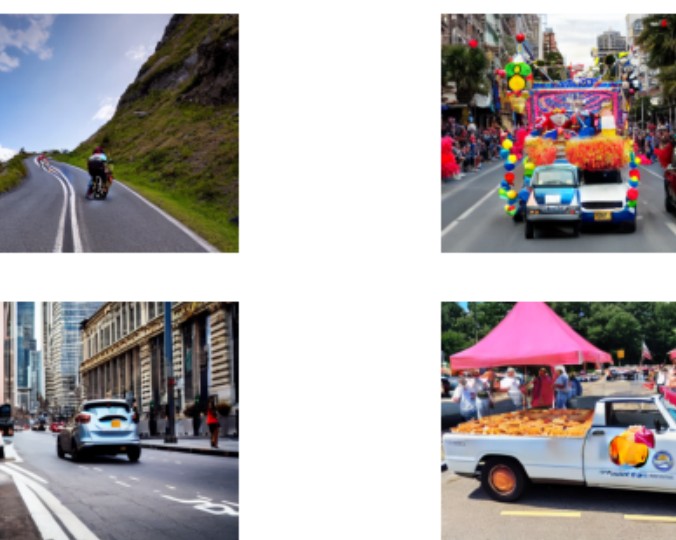

Figure 2: Generated images for the above random prompts from experiment 2b.

