# OpenReview forum: "Statistical Property Testing for Generative Models"
_ICLR.cc/2023/TinyPapers — Submitted to Tiny Papers @ ICLR 2023_

### Official Review · Reviewer_E28Y · 2023-03-29

**Confidence:** 3

**Summary Of Contributions:**

The authors introduce a framework for statistically verifying whether data produced by generative models satisfy certain properties with a given confidence level. This is essential for ensuring the suitability of synthetic data-points for specific use cases, such as model training. The proposed methodology is applied to both standard image and image/text generative models.

**Rating:**

High Potential (HP): a submission which meets the reviewing criteria and has potential to make an impact on the field

**Strengths And Weaknesses:**

Strengths:

1. The framework is applicable to various types of generative models, including those producing image and image/text data.

2. The statistical approach allows users to set a desired confidence level for the properties to be satisfied by the generated data.

3. The work contributes to a better understanding of synthetic data suitability and its implications for model training and other applications.

**Suggested Changes:**

None

---

### Official Review · Reviewer_dBtr · 2023-03-30

**Confidence:** 5

**Summary Of Contributions:**

In this work, the authors present a simple framework for statistically checking if the data produced by a generative model satisfy some property with a given confidence level. They apply their methodology on standard image, as well as on image/text generative models.

**Rating:**

Clear, Correct, and Reproducible (CCR): a submission which meets the reviewing criteria

**Strengths And Weaknesses:**

## Strengths

- The submission presents a simple framework for statistically checking if the data produced by a generative model satisfy some property with a given confidence level.
- The methodology is presented in a clear and concise way, and the experiments show that the proposed method can effectively verify whether the generated data satisfy the desired properties.
- The paper has communicated the methods clearly.

## Weaknesses

- The submission's method is very naive and relies on training a simple classifier. This approach has been used in various other studies, and there are no significant advancements or novel contributions in this submission. Without novelty, the contribution of the study may be limited.
-The submission only includes a limited number of experiments, which may not be sufficient to draw comprehensive conclusions. The experiments conducted may not cover all possible scenarios, and this may limit the generalizability of the results. Further experimentation and analysis are necessary to confirm the validity of the findings.
- An in-depth study in this area was done by Betzalel et al., 2022. The paper does contrast this with some other unrelated approaches in a similar broad area but does not particularly contrast their approach with other approaches in this area and I find this work to be not well compared with other approaches.

## References

Betzalel, Eyal, et al. "A Study on the Evaluation of Generative Models." arXiv preprint arXiv:2206.10935 (2022).

**Suggested Changes:**

- The introduction could be expanded to provide more context and background information about the problem the authors are trying to solve. The authors should also clarify the novelty of their approach and how it differs from previous work in the field.
- The authors should expand the scope of their experiments to include a larger and more diverse set of datasets
- Resize Table 1 to fit `\textwidth`
- Minor mistakes in writing

> We want to test the hypothesis that G outputs data that satisfy the property with sufficiently
high probability, e.g. we formulate the (null) hypothesis H0 = {p ≤ p0}, where p =
Pmathbfx∼G[ϕ(x) = 1] is the probability that G’s outputs satisfy the property, and p0 is a required
lower bound that we want to achieve. The goal is then to verify that H0 holds with sufficiently high
confidence. From Hoeffding’s inequality, we can easily derive the following:

---

### Public Comment · ~Bhavesh_Neekhra1 · 2023-04-10
**Very interesting work**

This work primarily works with image dataset. Is there similar work for synthetically generated tabular data?

---

### Comment · Area_Chair_mNs4 · 2023-06-01
**Archival**

This work meets the threshold for archival, contents the URM statement and is deanonymized

---

### Meta-Review · Area_Chair_mNs4 · 2023-04-02

**Recommendation:** Invite to present
**Confidence:** 4

**Metareview:**

The method is simple and can potentially be used for better understanding synthetic data from generative models. The method and the experiments are mostly clearly presented. But some of the reviewers are concerned with the novelty, lack of experiments, and comparison with related works. Another limitation is that this paper uses neural networks for checking the properties, which does not provide formal guarantees by itself and can introduce errors that are not considered in the paper.


**Summary:**

This paper presents statistical tests on whether a generative model generates data satisfying some property at a given confidence level.

**Reason For Not Giving A Higher Recommendation:**

This paper lacks comparison with existing evaluation works. Also, using neural networks for checking the properties can introduce errors in the statistical tests, which is not considered.


**Reason For Not Giving A Lower Recommendation:**

The presentation is mostly clear and mostly sound other than the aspects mentioned above.

---

### Decision · Program_Chairs · 2023-04-07

Invite to present